# Exploration of an Inflection Point of Ventilation Parameters with Anaerobic Threshold Using Strucchange

**DOI:** 10.3390/s22072682

**Published:** 2022-03-31

**Authors:** Takenori Aida, Akira Shionoya, Hirofumi Nonaka, Kouji Hayami, Hisashi Uchiyama, Masahiro Nagamori, Satoshi Ohhashi, Mai Kobayashi, Tsugumi Takayama, Shinji Kimura

**Affiliations:** 1Integrated Bioscience and Technology Major, Graduate Schools of Engineering, Nagaoka University of Technology, Nagaoka-shi 940-2188, Japan; s165042@stn.nagaokaut.ac.jp (T.A.); nonaka@kjs.nagaokaut.ac.jp (H.N.); utiyama@vos.nagaokaut.ac.jp (H.U.); nagamori@kjs.nagaokaut.ac.jp (M.N.); ohashi@vos.nagaokaut.ac.jp (S.O.); 2Data Dock Ltd., Nagaoka-shi 940-1140, Japan; hayami@datadock.co.jp; 3Department of Physical Threrapy, Seiryou, Junior College of Rehabilitation, Nagaoka-shi 940-2138, Japan; maikobayashi@seiryou-reha.ac.jp; 4Rehabilitation Center, Niigata University Medical and Dental Hospital, Niigata-shi 951-8520, Japan; takayama-c@med.niigata-u.ac.jp (T.T.); skimura@med.niigata-u.ac.jp (S.K.)

**Keywords:** anaerobic threshold, inflection-point exploration, strucchange, ventilation expired gas volume, excess CO_2_ production

## Abstract

(1) Background: When measuring anaerobic work threshold (AT), the conventional V-slope method includes the subjectivity of the examiner, which cannot be eliminated completely. Therefore, we implemented an engineering method using strucchange to objectively search for the inflection point of AT. (2) Methods: Seventeen subjects (15 men and 2 women) were included in the study. The subjects rode an ergometer and performed a ramp load test for 18 min and 30 s. (3) Results: In VE (Ventilation), 11 out of 12 subjects had the same results with 95% confidence intervals for the AT by the strucchange and respiratory metabolic apparatus. In VCO_2_ (Carbon dioxide emissions), 9 out of 12 subjects had the same results with 95% confidence intervals for the AT with the strucchange and respiratory metabolic apparatus. In VE, 3 out of 12 subjects showed the same results for respiratory metabolic analysis and the AT by the V-slope method. In VCO_2_, 3 out of 12 subjects showed the same results for the respiratory metabolic analysis and AT by the V-slope method in VCO_2_. (4) Conclusions: Strucchange was more objective and significant in identifying the AT than the V-slope method.

## 1. Introduction

In recent years, the average life expectancy in Japan has been nearing 100 years. Therefore, many researchers are focusing on the relationship between long life expectancy and good health throughout one’s life. In other words, studies on maintaining good health without compromising the quality of life (QOL) have been gaining popularity. However, good health and QOL are directly related to individual awareness. Although the current coronavirus pandemic is forcing people to restrict their daily activities, awareness regarding the importance of individual health is being raised by making people aware of the importance of wearing masks, hand sanitization, social distancing, vaccination, and other measures. The risks of lifestyle-related diseases, such as cancer, diabetes, cardiovascular diseases, as well as age-related decline in physical functions, are known to be reduced with appropriate physical activity, especially indoor exercises to relieve the stress caused by the lack of physical activity. Toshima et al. [1] reported that heart disease has been the second leading cause of death since 1985 due to dietary changes. In addition, ischemic heart disease (IHD) has become the leading cause of death worldwide [2]. Wannamethee et al. [3] reported that in addition to preventing cardiac disease, maintaining the quality of life (QOL) after cardiac disease leads to an increased healthy life expectancy. Therefore, cardiac rehabilitation exercise therapy is essential for secondary prevention in patients with a history of heart disease [4]. Although excessive exercise load is contraindicated, an appropriate exercise index is used in rehabilitation exercise therapy. An optimal exercise load intensity index called the anaerobic threshold [5] (AT, also known as anaerobic work threshold) considers safety and efficiency. The AT reflects exercise intensity and O_2_ uptake immediately prior to anaerobic metabolism occurring alongside aerobic metabolism during load-incremental exercise when the associated change in gas-exchange parameters occurs. Many studies on exercise therapy that use an appropriate exercise load index have been conducted on the elderly and cancer patients [6,7]. Tanebe et al. [8] reported that exercises that do not exceed the AT do not cause an excessive elevation of catecholamines in the blood and are safe exercises. There are two methods for determining the AT, namely the ventilatory threshold (the aforementioned ventilatory gas analysis) and lactate threshold (the lactate concentration in blood) methods. The former is non-invasive for the examinee and is the standard method used [9]. However, the ventilatory threshold is not suitable as an indicator of endurance training. The latter method is invasive, requires measurement techniques using sophisticated and expensive equipment, and relies on the experience of physicians and healthcare professionals [10]. In a previous study on AT determination, Tsubusadani et al. [11] described a method called cardiac work threshold, which determines the AT based on non-invasive data such as heart rate; however, this method requires improvement in measurement accuracy due to the effect of respiration on heart rate variability. Takemura et al. [12] proposed a technique to control respiratory rate during progressive load exercise using a bicycle ergometer, such as locomotor–respiratory coupling (LRC). Kimoto et al. [13] found that the maximum intensity spectrum of the frequency analyzed respiratory curve (corresponding to the respiratory rate) was significantly lower than that of the point at which the transition from low-frequency to high-frequency components was determined and used these intensity values as the exercise intensities before and after AT. The V-slope method [14,15], a conventional analog (Figure 1), is also used as an AT identification method. This method plots the change in carbon dioxide emissions (VCO_2_) against oxygen uptake (VO_2_) in exhaled air. However, the extraction of structural change points [16] includes subjectivity by the examiner and cannot be objectively identified. In other words, a method to objectively determine or identify the AT has not yet been established. Against this background, the purpose of this study was to develop a more objective method of searching for inflection points of the AT using strucchange [17] for the change points of each ventilation index in exhaled gas during exercise loading. In Wasserman’s definition [18], the AT is defined based on each ventilation index (Figure 2) [19]. An explanation of the definition based on each ventilation index is as follows. (1) Nonlinear increase in ventilation (VE: L/min) with increasing exercise intensity; (2) nonlinear increase in VCO_2_ (ml/kg/min); (3) nonlinear increase in respiratory quotient RQ; (4) increase in oxygen equivalent VE/VO_2_ without a change in carbon dioxide equivalent VE/VCO_2_, and (5) increase in end-expiratory oxygen concentration (PETO_2_: mmHg) without a change in end-expiratory carbon dioxide concentration (PETCO_2_: mmHg).

## 2. Materials and Methods

### 2.1. Subjects

A total of 17 subjects (age 21 ± 1.4 years; BMI 21 ± 1.9), 15 healthy men and 2 healthy women, engaged in regular daily exercise were included in the study. Informed consent was obtained from all the subjects involved in the study, which was conducted according to the guidelines of the Helsinki Declaration. This study was approved by the ethical review of the Nagaoka University of Technology (project identification code: H29-4, approval date: 12 May 2017). The experiments were conducted in the sports engineering laboratory of the Nagaoka University of Technology and the laboratory of the Seiryou Rehabilitation Institute. Table 1 lists the BMI and smoking history of the subjects.

### 2.2. Experimental Protocol and Setup

The experimental protocol and apparatus are presented in Figure 3 and Figure 4, respectively. After adjusting the saddle height, the subject rode a bicycle ergometer (AEROBIKE EZ101, COMBIWELLNESS Corp., Konami Sports Co., Ltd., Tokyo, Japan). Both pedals were fixed, and they wore an expiratory gas mask. To stabilize the subject’s heart rate, the subject was kept in a resting state (Rest) on the bicycle for 3 min and then warmed up for 3 min at a constant pedal rotation speed of 50 rpm on the ergometer (baseline load: 20 W). The load level (10 W) was increased every 30 s, and the pedal speed was maintained at 50 rpm. The duration of the exercise was 9 min and 30 s. After the exercise, there was a cool-down period of three minutes before the termination of the experiment. The total time from rest to the end of the experiment was 18 min and 30 s. The criterion for discontinuation of the experiment was when the heart rate became higher than the subject’s maximum heart rate (220 − age (beats/min), where “age” is the age of the subject) [20], or when the subject complained of physical discomfort. The respiratory metabolic measurement device was Vmax 29 s, manufactured by Sensor Medics. The analysis was performed using the breath-by-breath method [21].

### 2.3. Experimental Analysis

Each ventilation index was recorded on a PC, converted to A/D, averaged for 10 s, and smoothed with a simple moving average (3 points). The smoothed data were saved on a USB drive and analyzed using the strucchange package in R language on another PC. The AT of each subject was calculated using the analysis function (V-slope method) of the respiratory metabolic measurement device (Vmax 29 s) and reflected as a solid vertical line in the graph (time axis) of each ventilation index. Furthermore, the AT identification by the V-slope method, which is an analog method, was performed by two examiners: one of the authors (examiner A) and a clinical laboratory technician with more than 20 years of clinical experience (examiner B). Each examiner performed the V-slope method twice, and the mean value (converted to W) was used. Interclass correlation coefficients (ICCs) [22] were calculated using Excel 2016, assessing the reliability.

## 3. Results

### 3.1. Results of V-Slope Method and Strucchange and Respiratory Metabolic Analysis

In Figure 5, the AT by the respiratory metabolic apparatus, strucchange, and V-slope method are shown as blue circles, red squares, and black circles, respectively.

### 3.2. Results from Strucchange and Respiratory Metabolic Analysis

The strucchange results for the VE of subject K are shown in Figure 6. The horizontal axis shows the number of breakpoints, the left vertical axis shows the BIC analysis value, and the right vertical axis shows the RSS analysis value. The number of breakpoints is the minimum value for both BIC and RSS, indicating three breakpoints based on strucchange.

The results of the strucchange results of VE (subject K) after smoothing are shown in Figure 7. The vertical solid line shows the AT by the respiratory metabolic apparatus. The vertical dashed line indicates the change point by strucchange, and the red H-shape indicates the confidence interval of strucchange. The confidence intervals were calculated to be within 95%.

The results of strucchange in the VCO_2_ of subject K are shown in Figure 8. Because the number of change points is minimal for both BIC and RSS analyses, there are four change points by strucchange.

The results of strucchange results of VCO_2_ (subject K) after smoothing are shown in Figure 9. The vertical solid line shows the AT by the respiratory metabolic apparatus. The vertical dashed line indicates the change point by strucchange, and the red H-shape indicates the confidence interval of strucchange. The confidence intervals were calculated to be within 95%.

The number of breakpoints (RQ). The results of strucchange in the VCO_2_ of subject K are shown in Figure 10. Because the number of change points is analyzed through both BIC and RSS analyses, there are three change points detected by strucchange.

The results of strucchange results of RQ (subject K) after smoothing are shown in Figure 11. The vertical solid line shows the AT by the respiratory metabolic apparatus. The vertical dashed line indicates the change point by strucchange, and the red H-shape indicates the confidence interval of strucchange. The confidence intervals were calculated to be within 95%.

## 4. Discussion

### 4.1. Comparison between Strucchange and Respiratory Metabolic Analysis

As shown in Figure 7, Figure 9 and Figure 11, strucchange was used to detect the points of change in the VE, VCO_2_, and RQ associated with a gradual increase in exercise load, and the points of change in all 17 subjects (A–Q) were detected. This is a result of the structural change extraction by strucchange. The background of the extraction of the structural change point is considered to be the process of the increase in CO_2_ storage in the tissue [16], starting from the buffering by HCO_3_^−^ against the increase of CO_2_ in the blood during the gradual increase in exercise load. One hundred watt AT detection from Figure 7, and one hundred five watt AT detection from Figure 9. The AT detection by strucchange was close to each other from the two results. In other subjects, Table 2 shows that 11 out of 12 subjects (B–L) had the same results with 95% confidence intervals for the AT by the strucchange and respiratory metabolic apparatus (excluding those who could not be analyzed). Table 3 shows that 9 out of 12 subjects (B–H, J, and K) had the same results with 95% confidence intervals for the AT with strucchange and respiratory metabolic apparatus (excluding those who could not be analyzed). Table 4 shows that 2 out of the 17 subjects (B,H) had the same results, that is, 95% confidence intervals for the AT with strucchange and respiratory metabolic apparatus (excluding those who could not be analyzed). Compared with the aforementioned results for VE and VCO_2_, the results showed a significant decrease in the number of people. The respiratory quotient, RQ, is expressed as VCO_2_/VO_2_ and is defined as a measure of carbon dioxide emissions divided by oxygen uptake. As described previously, VCO_2_ or VO_2_ can be analyzed separately; however, when they are analyzed as a combined parameter, the results differ significantly due to the inclusion of errors from the analysis of each parameter. The cause of this difference has not yet been identified. Therefore, the RQ results require further investigation. There are two reasons why the respiratory metabolic analysis system could not detect the change points. One reason is that the subject could not continue the experiment and became exhausted. The other reason is the consideration of individual physiological effects on the subject. In the latter case, it is possible that the conventional V-slope method [16] could not detect this effect. As Nishijima et al. [23,24] pointed out, it is possible that VCO_2_ was shifted to the right in the V-slope method during gradual exercise, and the AT could not be detected. This may be due to the state in which increased CO_2_ in the blood is not sufficiently discharged from the alveoli during exercise load escalation; that is, the chemoreceptors are not functioning. In other words, CO_2_ gas is stored in the alveoli and is not sufficiently expelled from the alveoli. Therefore, it is possible that the CO_2_ that is no longer expelled is absorbed by the alveoli and circulated back into the bloodstream, causing a rightward shift (Figure 12) [23]. First, the conventional V-slope method is based on the complete decomposition of glucose C_6_H_12_O_6_ + 6O_2_ → 6CO_2_ + 6H_2_O. It is difficult to say that the AT can be identified theoretically. Second, the detection algorithm used in the respiratory metabolic apparatus was based on the V-slope method. As long as the slope of the increase in VCO_2_ relative to VO_2_ is assumed to be greater than θ = 45°, it is challenging to detect AT, as shown in the experimental results. This result suggests that AT detection is insufficient in respiratory metabolic analysis based on the V-slope method, whereas that based on strucchange, which extracts structural change points, is significant.

### 4.2. Comparison of the V-Slope Method with Strucchange and Respiratory Metabolic Analysis

Table 2 shows that 4 out of 17 subjects (A, B, F, J) had the same results with 95% confidence intervals for VE by the strucchange and V-slope methods. Similarly, 3 of 12 subjects (B, F, J) showed the same results for the AT by respiratory metabolic analysis (excluding those who could not be analyzed) and the V-slope method. Table 3 shows that 5 out of 17 subjects (A, B, F, J, Q) had the same results (95% confidence intervals) for VCO_2_ by the strucchange and V-slope methods. Similarly, 3 out of 12 subjects (B, F, J) showed the same results in respiratory metabolic analysis (excluding those who could not be analyzed) and the AT by the V-slope method. Table 4 shows that 1 out of 17 subjects (B) had the same results with 95% confidence intervals for RQ by the strucchange and V-slope methods. This result indicates a difference in the number of patients between the V-slope method and the statistical analysis methods (strucchange and respiratory metabolic analysis). Using the V-slope method, the intraclass correlation coefficient ICC (2, 2) between the two examiners was 0.4449. According to Landis et al.’s criteria, the confidence coefficient was moderate (0.41–0.60) [25]. With this result, the judgment using the V-slope method is acceptable. However, the ICC values are only point estimates, and the theoretical basis is unclear. However, Shrout et al. [22] recommended presenting a confidence interval (lower limit). The lower limit was 100% (1-α) (α: significance level). Here, if the significance level α = 0.1, the confidence interval (lower limit) is 90%, and the above ICC is not satisfied. The V-slope method is considered to have a limitation in AT identification because of individual errors and negligence errors caused by each examiner’s subjectivity, compared with the analysis process described above in Section 4.1. To solve this problem in the V-slope method, it is necessary to improve the quality of judgment by increasing the number of examiners’ tests, reviewing the judgment criteria, and increasing the number of examiners. As this study focuses on AT search, detailed estimation of sample size is not performed. The comparison of the V-slope method and the strucchange and respiratory metabolic apparatus showed that the V-slope method has low reproducibility in the statistical analysis of ICC values. It is necessary to consider systematic errors when making judgments, which suggests the superiority of statistical methods.

## 5. Conclusions

In this study, we performed a progressive loading test using a bicycle ergometer, measured and analyzed each ventilation parameter using a respiratory metabolic device, searched for the change point of the ventilation parameter by strucchange, and conducted an anaerobic threshold (AT) search. This study is summarized as follows.

(1)It was possible to detect the change point by strucchange in each ventilation index (VE, VCO_2_).(2)We confirmed that the results of respiratory metabolic analysis and the confidence intervals by strucchange were almost identical in VE and VCO_2_. The RQ results differed significantly from the two aforementioned results due to possible errors in the analysis.(3)The V-slope method was objectively evaluated using inter-examiner reliability (ICC), but the results were unreliable.

However, the study also realized several methods and ways to improve the experimental procedure and the conclusion of the results. The first is to improve the quality of judgments by eliminating as many subjective factors as possible, such as the number of examiners in the V-slope method. The second is to make the AT detection system more versatile by considering past exercise habits and the presence or absence of personal medical history because of individual differences among subjects. Third, there is an error in analyzing RQ. It is necessary to fix the cause of the error and the analysis program. In the future, we would like to develop a medical-engineering collaboration system for diagnosis based on analysis using engineering methods and physiological ventilation indices.

## Figures and Tables

**Figure 1 sensors-22-02682-f001:**
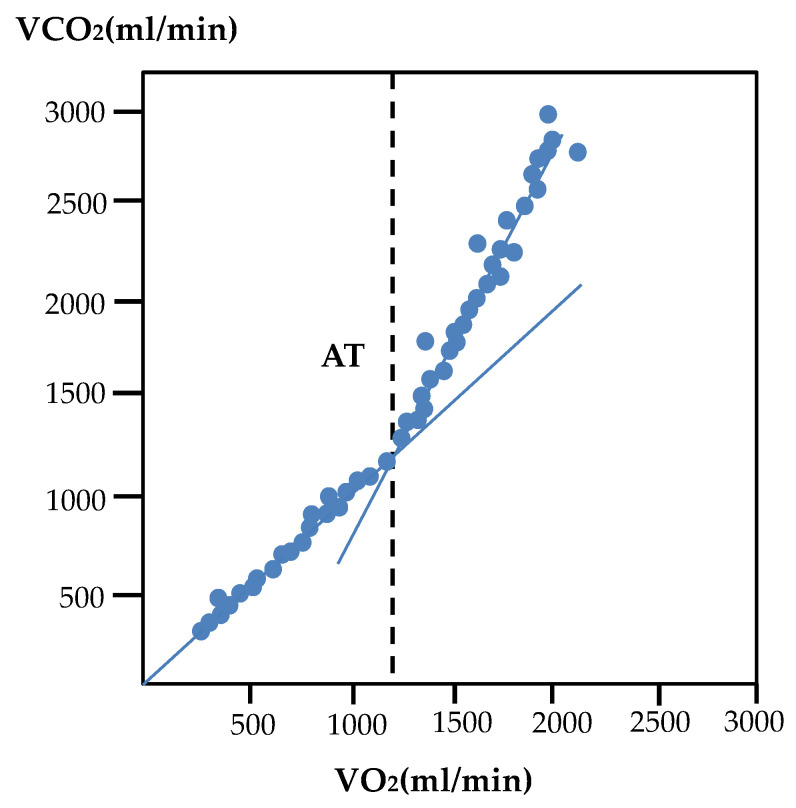
AT using the V-slope method. Eric et al. [14] presented a gas exchange plot from the submaximal exercise test of a representative participant to illustrate identification of the AT. With the V-slope method, the VCO_2_/VO_2_ plot is used to identify the point at which the VCO_2_ starts to increase more rapidly than VO_2_. The vertical dashed line represents the AT. Figure 1 was created based on subject data.

**Figure 2 sensors-22-02682-f002:**
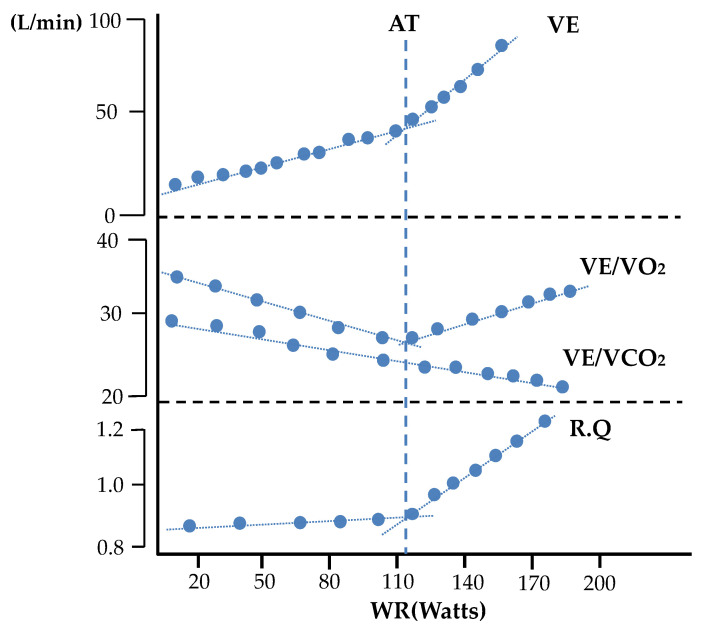
The AT is defined based on each ventilation index. Figure 2 was created by partially reorganizing the subject data.

**Figure 3 sensors-22-02682-f003:**
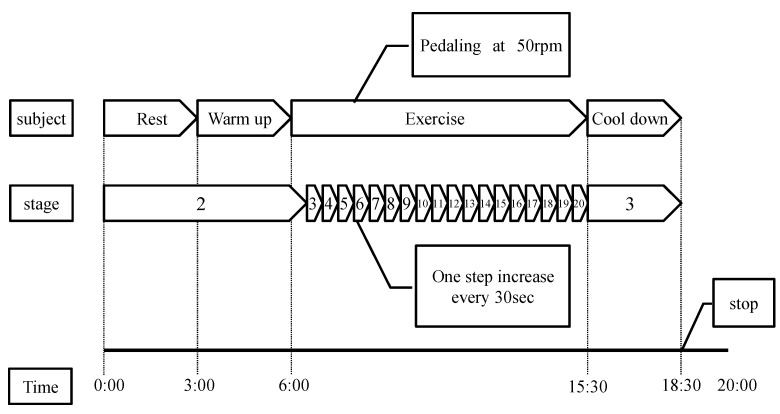
Experimental protocol.

**Figure 4 sensors-22-02682-f004:**
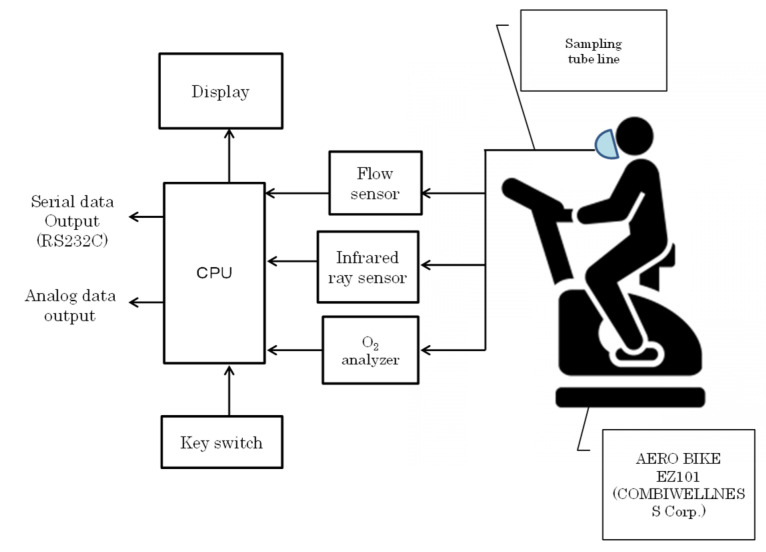
Experimental setup.

**Figure 5 sensors-22-02682-f005:**
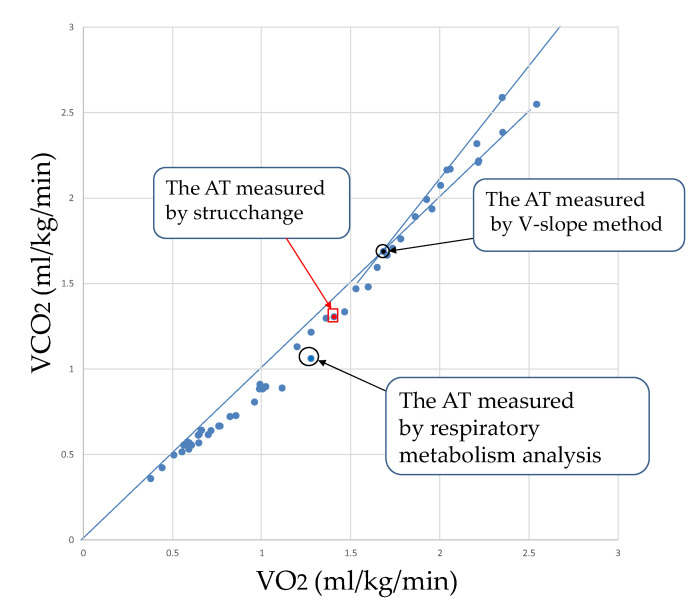
Comparison of each AT for subject K by the V-slope method, strucchange, and respiratory metabolism analysis.

**Figure 6 sensors-22-02682-f006:**
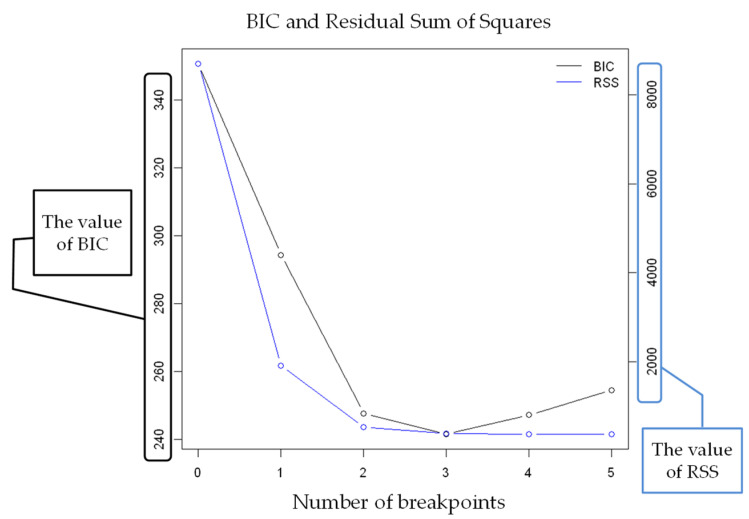
Number of breakpoints (VE).

**Figure 7 sensors-22-02682-f007:**
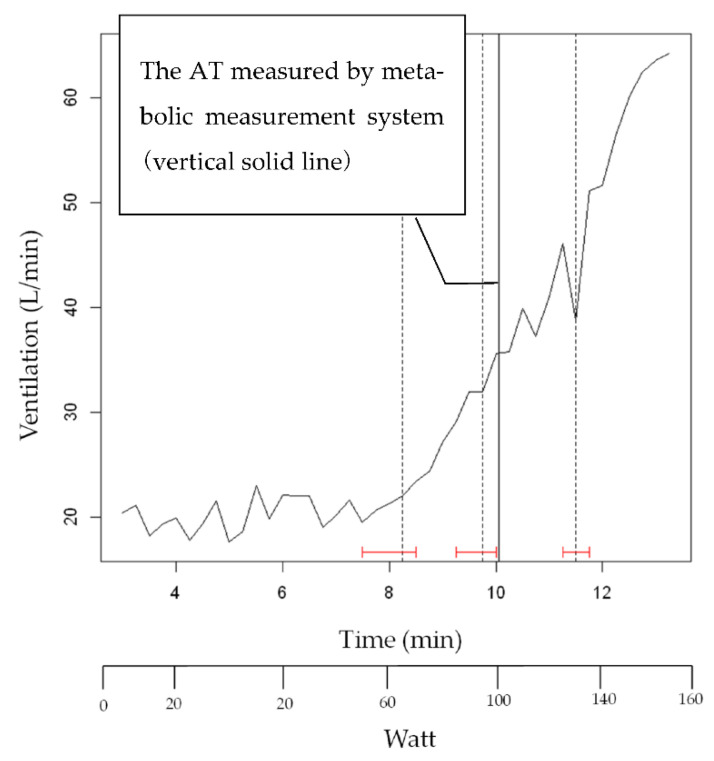
The results of the strucchange results of VE (subject K) after smoothing.

**Figure 8 sensors-22-02682-f008:**
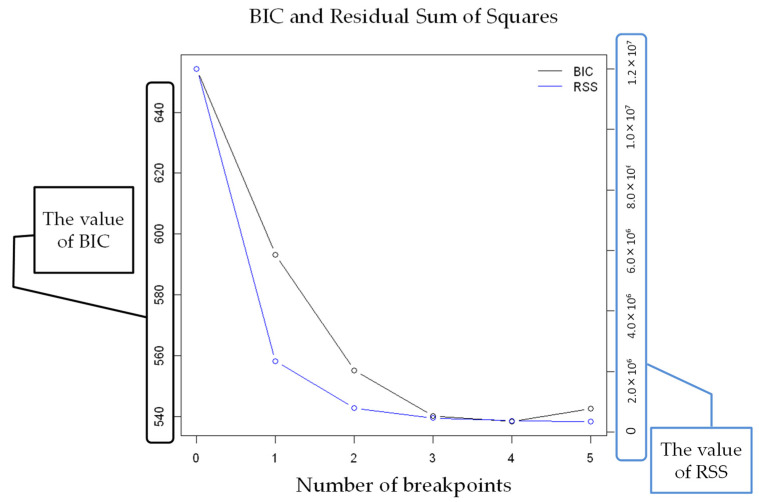
The number of breakpoints (VCO_2_).

**Figure 9 sensors-22-02682-f009:**
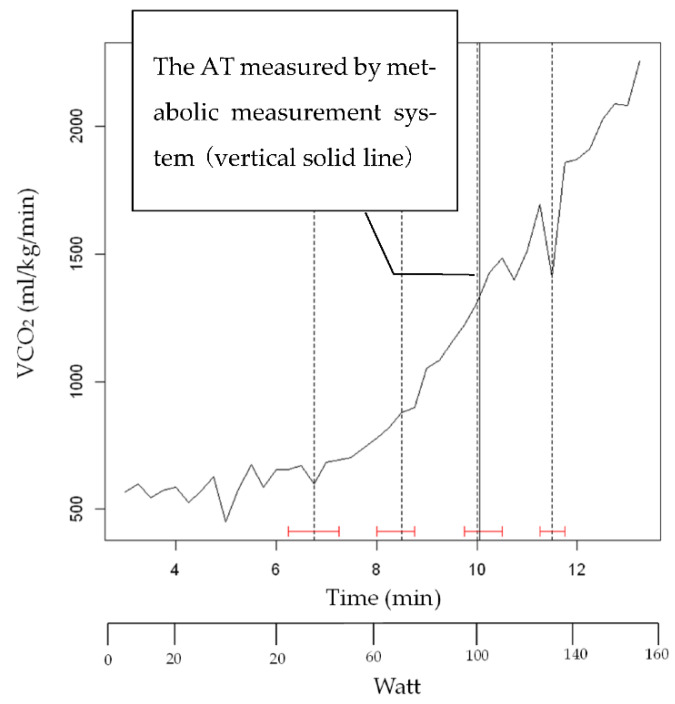
The results of strucchange results of VCO_2_ (subject K) after smoothing.

**Figure 10 sensors-22-02682-f010:**
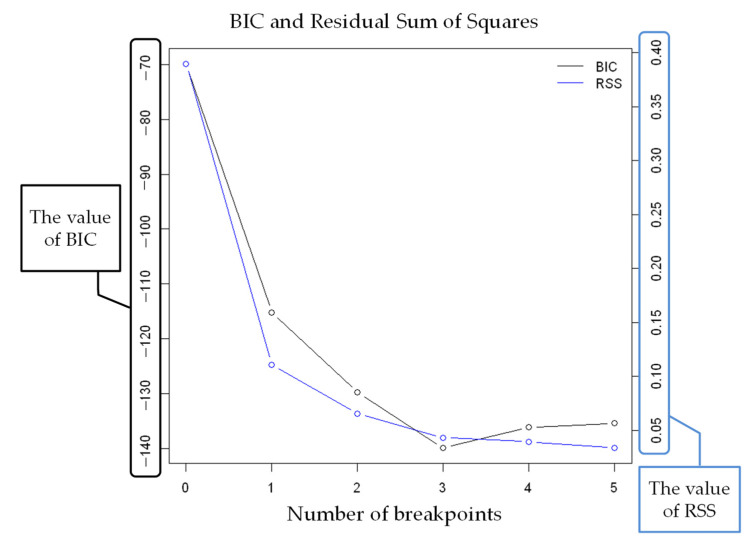
The number of breakpoints (RQ).

**Figure 11 sensors-22-02682-f011:**
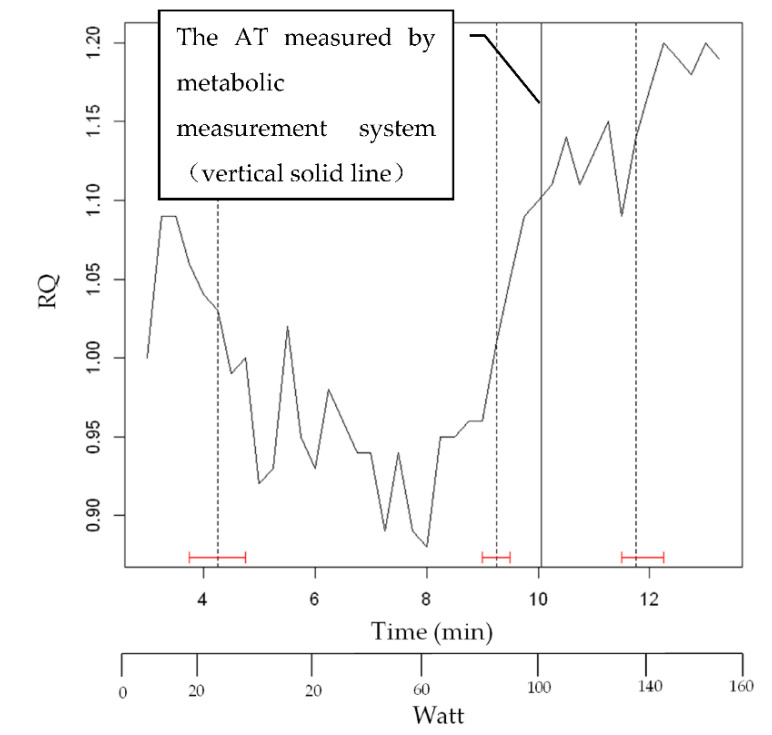
The results of strucchange results of RQ (subject K) after smoothing.

**Figure 12 sensors-22-02682-f012:**
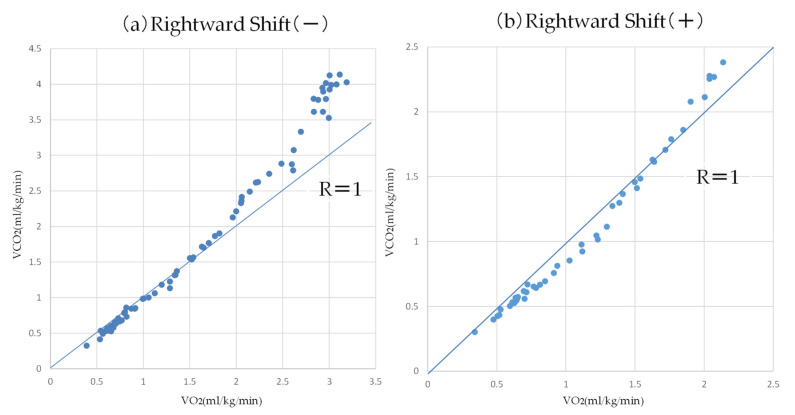
Typical examples of a V-slope without ((**a**): left) and with ((**b**): right) Rtshift. The shift of V-slope is judged relative to the R = 1 diagonal line [23]. Figure 12 was created based on subject data.

**Table 1 sensors-22-02682-t001:** BMI and smoking history of the subjects.

Subject	BMI	Smoking History
A	23.2	Non-smoker
B	19.8	Non-smoker
C	19.5	Non-smoker
D	19.7	Non-smoker
E	21.5	Non-smoker
F	22.8	Non-smoker
G	20	Non-smoker
H	24.9	Non-smoker
I	18.2	Non-smoker
J	21.8	Non-smoker
K	18.7	Non-smoker
L	21.4	Non-smoker
M	21.4	Non-smoker
N	18.2	Non-smoker
O	18.5	Non-smoker
P	19.8	Non-smoker
Q	20.5	Non-smoker
mean	21	-
SD	1.9	-

**Table 2 sensors-22-02682-t002:** AT by strucchange, V-slope methods, and respiratory metabolic analysis in VE (numbers in table converted to W).

Subject	Strucchange (95% CI)	Strucchange (Median)	Metabolic Measurement System	V-Slope Method
A	110	110	100	110
B	100–110	110	110	110
C	80–90	90	90	100
D	70–80	70	80	120
E	80–90	80	80	125
F	120–130	120	120	120
G	120–130	130	120	140
H	90–100	90	90	125
I	70–90	80	90	95
J	120–130	130	120	120
K	90–100	100	100	150
L	100	100	100	110
M	90–100	100	Not detected	85
N	100–110	105	Not detected	117.5
O	100	100	Not detected	112.5
P	80–100	90	Not detected	137.5
Q	100–110	100	Not detected	97
mean	-	100.3	100	116.1
SD	-	16.1	14.8	16.4

**Table 3 sensors-22-02682-t003:** AT by strucchange, V-slope methods, and respiratory metabolic analysis in VCO_2_ (numbers in table converted to W).

Subject	Strucchange (95% CI)	Strucchange (Median)	Metabolic Measurement System	V-Slope Method
A	110–120	110	100	110
B	110–130	120	110	110
C	90	90	90	100
D	70–80	80	80	120
E	80–90	80	80	125
F	120–130	130	120	120
G	120–130	130	120	140
H	90–100	90	90	125
I	100–110	110	90	95
J	120–130	130	120	120
K	100–115	105	100	150
L	120–130	125	100	110
M	100–115	105	Not detected	85
N	100–110	105	Not detected	117.5
O	90–100	95	Not detected	112.5
P	100–115	110	Not detected	137.5
Q	95–110	105	Not detected	97
mean	-	107.0	100	116.1
SD	-	15.9	14.8	16.4

**Table 4 sensors-22-02682-t004:** AT by strucchange, V-slope methods, and respiratory metabolic analysis in RQ (numbers in table converted to W).

Subject	Strucchange (95% CI)	Strucchange (Median)	Metabolic Measurement System	V-Slope Method
A	110–120	110	100	110
B	110–120	120	110	110
C	100–110	110	90	100
D	90–100	100	80	120
E	100–110	100	80	125
F	100–110	110	120	120
G	80–90	90	120	140
H	60–100	70	90	125
I	70–80	70	90	95
J	100–110	110	120	120
K	70–80	70	100	150
L	50–65	55	100	110
M	60–75	65	No detected	85
N	20	20	No detected	117.5
O	75–85	80	No detected	112.5
P	70–80	75	No detected	137.5
Q	50–75	60	No detected	97
mean	-	83.2	100	116.1
SD	-	24.8	14.8	16.4

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
