# Peer review of "Exploration of an Inflection Point of Ventilation Parameters with Anaerobic Threshold Using Strucchange"

_sensors, 2022, doi:10.3390/s22072682_

Round 1

Reviewer 1 Report

Very interesting and well-written study.

I have some comments:

  • Please add a table with patients' baseline characteristics, including BMI and smoking status.
  • Did authors estimate sample size? Please add in data analysis section
  • Please add ethics protocol approval number
  • I found some minor English errors throughout the paper. Please have a re-check.

Reviewer 2 Report

Dear Authors:

I appreciate the possibility to review the paper entitled "Exploration of an inflection point of ventilation parameters with anaerobic threshold using strucchange" by the authors Aida et al. In this paper, using a maximal exercise test on a cycle ergometer, it is proposed to determine the anaerobic threshold using the strucchange analysis of the time series and compare it with the V slope method and the respiratory and metabolic indexes proposed by Wasserman. In this respect, I find the work interesting, well-structured, and with promising results. However, I would like to make a few comments.

-The paper's aim should be the same in the abstract and the introduction.

- I suggest presenting the paper's aim as the last idea in the introduction to make the manuscript easier to read.

- Given the poor quality of the images (Fig 1 and 2), it might be better to redraw them and describe them as an adaptation of the original publication.

- The methods do not specify how the baseline load of the participants was determined.

- In figure 3, the indication of the test time appears in the reverse direction. Time "0" should be at the start.

- I recommend deleting graphs 5, 8, and 11 because graphs 7, 10, and 13 show the same information together with lines obtained by analyzing the structural change of the time series.

Author Response

Response 1:  I described the paper's aim.

Response 2:  I redraw Figures1-2 and described them as an adaptation of the original publication.

Response 3:I described baseline load of the participants.

Response 4:I revised Fig3.

Response 5:I deleted Fig5,8,11.

This manuscript is a resubmission of an earlier submission. The following is a list of the peer review reports and author responses from that submission.